# Isolating single cycles of neural oscillations in population spiking

**Ehsan Sabri** [ID][1¤*], **Renata Batista-Brito**[1,2,3*]

**1** Dominick P. Purpura Department of Neuroscience, Albert Einstein College of Medicine, Bronx, New York, United States of America, **2** Department of Psychiatry and Behavioral Sciences, Albert Einstein College of Medicine, Bronx, New York, United States of America, **3** Department of Genetics, Albert Einstein College of Medicine, Bronx, New York, United States of America

¤ Current address: Department of Psychiatry, Columbia University, New York, New York, United States of America

* ehsan.sabri@gmail.com (ES); renata.brito@einsteinmed.edu (RB-B)

**Data availability statement:** Data and code are available at https://figshare.com/articles/dataset/isoCycle_Paper_Data_and_Code/28330619/1. Data regarding Figs 3 and 5 are

## Abstract

Neural oscillations are prominent features of brain activity, characterized by frequency-specific power changes in electroencephalograms (EEG) and local field potentials (LFP). These oscillations also appear as rhythmic coherence across brain regions. While the identification of oscillations has primarily relied on EEG and LFP, they are also present in neuronal spiking. However, several questions remain unanswered: How do spiking oscillations relate to field potential oscillations? How are spiking oscillations correlated across brain regions? And how are they connected to other physiological and behavioral measures. In this study, we explore the potential to detect individual cycles of neural rhythms solely through the spiking activity of neurons, leveraging recent advances in the high-density recording of large neuronal populations within local networks. The pooled spiking rate of many neurons within a local population reflects shared variation in the membrane potential of nearby neurons, allowing us to identify cyclic patterns. To achieve this, we utilize a Long Short Term Memory (LSTM) network, pre-trained on synthetic data, to effectively isolate and align individual cycles of neural oscillations in the spiking of a densely recorded population of neurons. We applied this approach to robustly isolate specific neural cycles across various brain regions in mice, covering a broad range of timescales, from gamma rhythms to ultra-slow rhythms lasting up to hundreds of seconds. These ultra-slow rhythms, often underrepresented in the LFP, were also detected in behavioral measures of arousal, such as pupil size and mouse facial motion. Interestingly, these rhythms showed delayed coherence with corresponding rhythms in the population spiking activity. Using these isolated neural cycles, we addressed two key questions: 1) How can we account for biological variation in neural signal transmission timing across trials during the sensory stimulation experiments? By isolating gamma cycles driven by sensory input, we achieved a more accurate trial alignment in the sensory stimulation experiments conducted in the primary visual cortex (V1) of mice. This alignment accounts for biological variability in sensory signal transmission times from the retina to V1 across trials, enabling a clearer understanding of neural dynamics in response to sensory stimuli. 2) How do spiking correlations across brain regions vary by timescale? We used

from https://janelia.figshare.com/articles/dataset/Eight-probe_Neuropixels_recordings_during_spontaneous_behaviors/7739750. The developed method is accessible in github https://github.com/esiabri/isoCycle. And can be applied on other datasets using this google colab notebook: https://colab.research.google.com/github/esiabri/isoCycle/blob/main/isoCycle_yourData_Colab.ipynb.

**Funding:** This work was supported by National Institutes of Health (NIH) grants R01EY034617, R21MH133097, R01EY034310 (to RBB), NARSAD Young Investigator Award (to RBB), Simons Bridge to Independence Award (to RBB) and Whitehall Award (to RBB). The funders had no role in study design, data collection and analysis, decision to publish, or preparation of the manuscript.

**Competing interests:** The authors have declared that no competing interests exist.

the distinct spiking cycles in simultaneously recorded brain regions to examine the correlation of spiking across brain regions, separately for different timescales. Our findings revealed that the delays in population spiking between brain regions vary depending on the brain regions involved and the timescale of the oscillations. This work demonstrates the utility of population spiking activity for isolating neural rhythms, providing insights into oscillatory dynamics across brain regions and their relationship to physiological and behavioral measures.

## Author summary

Neural oscillations, or rhythmic patterns of brain activity, have traditionally been studied by measuring electrical fields in the brain. In our research, we developed a novel approach that focuses directly on the activity of individual brain cells (neurons). Using machine learning, we created a tool to detect and analyze these rhythms, identifying individual cycles of brain rhythms across a wide range of timescales—from milliseconds to minutes. When we applied this method to recordings from mouse brains, we uncovered several noteworthy findings. First, we discovered that sensory information is processed in the brain as discrete packets of activity, rather than as a continuous stream. We also found that different brain regions synchronize their activity through these rhythms, with specific timing delays that depend on the rhythm's speed. Moreover, we found that the ultra-slow rhythms in brain activity are linked to behavioral cues like changes in pupil size and facial movements, indicating that these rhythms may reflect shifts in the animal's internal state. Our work sheds light on how the brain processes information and coordinates activity across its regions. It also introduces a powerful new tool exploring the fundamental mechanisms underlying brain function.

## Introduction

Neural rhythms, which represent distinct cyclic temporal patterns of activity in the network, are fundamental features of neural systems [1–6]. These oscillatory patterns in neuronal population activity are primarily detected by measuring adjacent electromagnetic fields [7]. Rhythmic activity typically arises within particular spectral bands (Fig 1A) as observed in the Local Field Potential (LFP) [8,9]. Individual neurons tend to fire spikes during specific phases of these rhythms [10–13] as revealed by methods that measure the coherency between the spiking activity and LFP [14,15]. This suggests that the probability of neuronal spiking is modulated by the same recurring temporal motifs—namely, cycles (Fig 1B), which underlie the spectral changes observed in LFP. The duration of these cycles is linked to their respective frequency bands (Fig 1B). For example, a gamma cycle spans approximately 20 milliseconds. Spectral analysis of neural signals, especially of LFP, has been the primary approach for linking rhythmic activities to various functions of the nervous system. Additionally, time-domain methods have been developed to detect repeating cycles in LFP [16,17]. However, the extent to which spiking activity contains information about the timing of individual cycles remains unclear. Identifying individual cycles as discrete neural events offers improved temporal resolution, enabling more precise alignment of these cycles and improved detection of their oscillatory bursts [16,18]. This enhanced temporal resolution is essential for investigating the

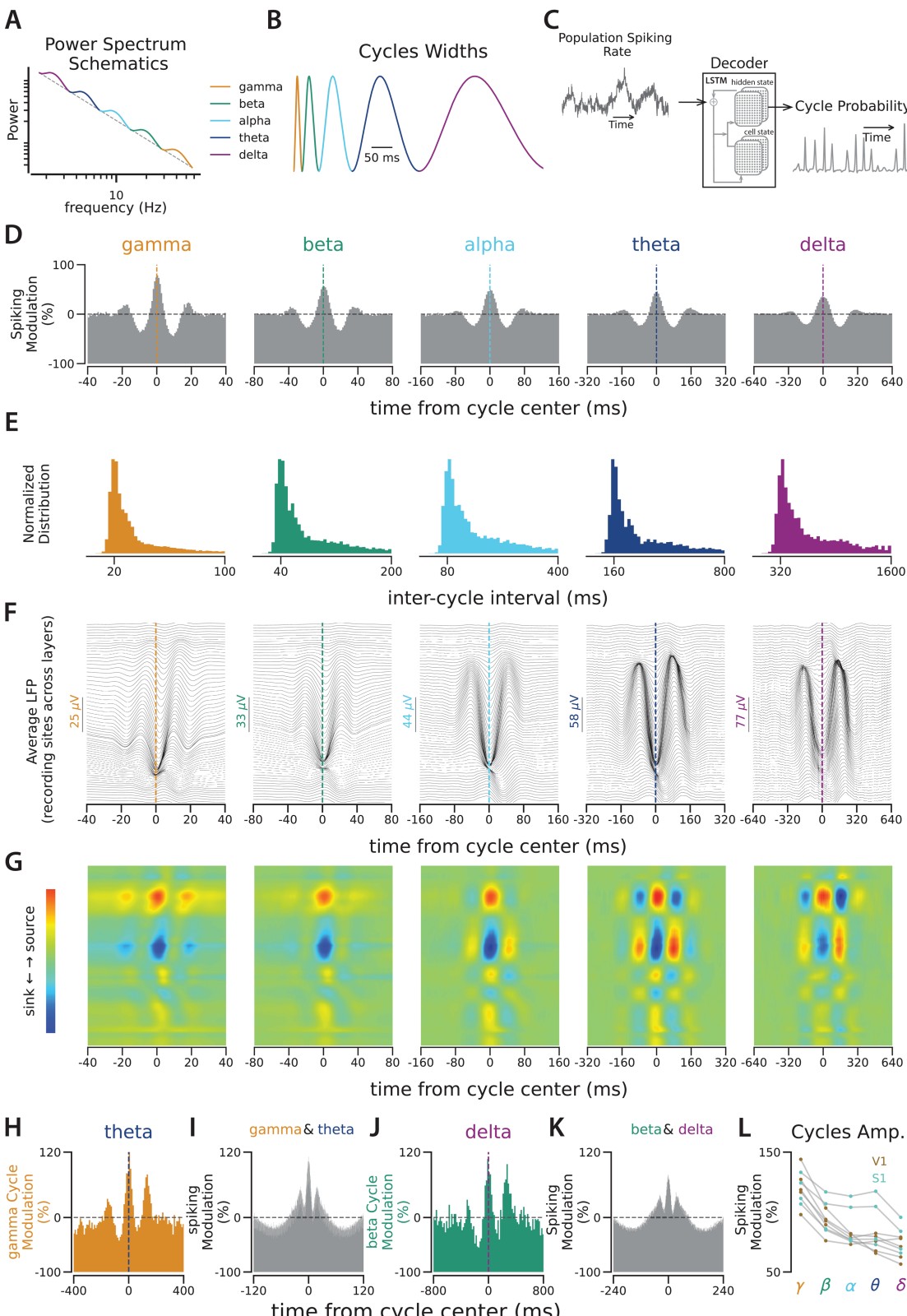

**Fig 1. Cycle Detection in Population Spiking. A.** Schematic illustrating the modulation of neural activity power spectrum across specific frequency bands commonly observed in neural recordings. The dashed line represents the 1/f power decrease in the spectrum, and the bumps represent the band-specific power modulations. Both axes are presented in logarithmic scale. **B.** Cycle width of rhythmic activities linked to spectral bands in A. **C.** The decoder evaluates the changes in network spiking over time and

returns the probability of the targeted cycle type at each time point. The decision for cycle detection is made based on the decoder output (see S1 Fig). **D.** Distribution of all population spike times aligned with detected cycle times for each cycle type during 6300 seconds of extra-cellular recording from 131 neurons in V1. 5000 cycle times were randomly chosen for each cycle type to generate this spiking distribution. The horizontal dashed line is the chance level (see S3 and S4 Figs for similar distributions around shifted and shuffled cycle times). The vertical line in each panel indicates the center of the detected cycles. The y-axis shows the amount of spiking modulation relative to the chance level. **E.** Normalized distribution of inter-cycle interval (ICI) for each cycle type, similar to panel D. **F.** Average LFP aligned with detected cycle times. Each panel corresponds to the cycle in the same column as in A. LFP was recorded with 64 channels across the width of the cortex. The upper traces correspond to channels in the superficial layers and the lower traces correspond to channels in deeper layers. Scales for each cycle type are adjusted for pattern visibility. **G.** Current source density (CSD) maps for the average LFP traces in F. **H.** Distribution of the gamma cycle times aligned with theta cycle times. **I.** Distribution of population spiking relative to co-occurrences of gamma and theta cycles. **J.** Distribution of the beta cycle times aligned with the delta cycle times. See S5 Fig for all pairwise relations of cycles. **K.** Distribution of population spiking relative to co-occurrences of beta and delta cycles. For spiking distributions relative to co-occurrences of other pairs of cycles, see S6 Fig. **L.** The Y-axis shows the trough-to-peak amplitude of the average shape of each cycle type, as in D, across 9 extracellular recordings: 5 recordings from V1 (brown) and 4 recordings from S1 (turquoise).

computational roles of these cyclic events within neuronal circuits and their correlations with behavior.

We hypothesize that individual cycles of rhythmic activity in neuronal spiking can be identified by analyzing the changes in the local population spike count. To achieve this we aggregated spikes from all single units within a densely recorded population of neurons and measured the population spike rate across time, treating it as a univariate signal [19,20]. In densely recorded extracellular activity, where tens of neurons are recorded simultaneously within a local network, variations in the rate of population spikes reflect shared fluctuations in sub-threshold activity across the network. Sub-threshold activity has also been shown to follow LFP [21,22]. Taken together, these observations suggest that patterns of cyclic neural activity can be identified by analyzing the pooled spiking rate of the recorded population. However, it should be noted that the rhythmicity in LFP and spiking activity are not always correlated, for example, beta-band activity in the primate motor cortex is dissociated between LFP and neuronal spiking [23,24].

To test this hypothesis, we employed Long Short Term Memory (LSTM) networks as decoders to isolate individual cycles of various neural rhythms in the spiking of neurons. We accomplished this by analyzing the variations in the population spike rate, pooling spikes from all recorded neurons within a local network, in the time domain. LSTM is a powerful tool for analyzing time-series data, capable of considering the history of the signal to identify learned events [25], which makes it well-suited for our investigation. We trained the decoder on a synthesized signal constructed using neural cycles as basis functions (Fig 1B), and incorporated noise and variations in the width of each cycle instance to match the spectral characteristics of the biological signal (Fig 1B–1C, S1 Fig, and Methods). Using the trained decoder, we then identified the timing of individual cycles underlying the fluctuations in the population spike counts during extracellular recording experiments. The decoder was trained on the synthetic signal that is matched to the spectral content of the biological signals. However, additional features not captured in the training signal could introduce bias when applying the decoder to real neuronal data. To assess the validity of the detected cycles in recorded data, we demonstrated that the average LFP around these cycles, identified in population spiking, exhibits similar cyclic patterns. We also confirmed known neural cross-frequency couplings using these isolated cycles in population spiking.

Using this approach, we successfully isolated individual cycles of distinct rhythms across multiple regions within the mouse brain. In the primary visual and somatosensory cortices (V1 and S1) of mice, we effectively isolated cycles with durations ranging from gamma

(around 20 ms) to several hundred seconds. Furthermore, we applied this method to simultaneous recordings from tens of regions in the mouse brain [26]. We used the isolated cycles in the population spiking to investigate two outstanding questions in the field: 1) In visual stimulation experiments, we used isolated sensory-driven gamma cycles in V1 to improve trial alignment precision. This realignment compensated for variations in signal transmission times from the retina to V1 across trials. 2) In simultaneous recordings from multiple regions in the mouse brain, we used the isolated cycles in each time scale to measure the correlated spiking and we observed specific correlation depending on the brain regions and time scales.

## Results

### Cycle detection in population spiking of mouse primary sensory cortex

We developed a nonparametric, time-domain method to isolate individual cycles of neural oscillations from population spiking. For this cycle detection, we employed an LSTM (Long Short-Term Memory) network, functioning as a decoder (Fig 1C). This network was trained on synthesized signals constructed with known neural cycles, such as gamma, beta, and others, and we validated the decoder's performance using generated ground truth data (S1 Fig, Fig 1B–1C, and Methods). This method works significantly better than other methods that are based on the spectral analysis (S2 Fig), because during the training on the synthesized data, the decoder has learned to handle the variable delays between consecutive cycles as well as the jitter on the width of each cycle (S1 Fig). However, the assumption that the neural signal can be replicated using just the known neural cycles as the basis functions may limit the method's applicability. To test the cycle detection in neuronal spiking, we recorded extracellular activity using single-shank linear silicon probes along the width of the primary visual cortex (V1) and the primary somatosensory cortex (S1) of head-fixed mice. In each recording session, we combined the spike times from all recorded neurons (n=55±31, mean±std across nine recording sessions) to identify the timing of individual cycles of the rhythms ranging from gamma to delta (Fig 1C, S1 Fig, and Methods). We then used the times of the detected cycles for each cycle type to calculate the distribution of population spiking around each cycle type. These distributions represent the average shape of each cycle, as reflected in the spiking activity of the network (Fig 1D). To compare cycles across recordings and time scales, we normalized the average cycle shapes relative to the chance level, which is the average spiking during the recording (see S3 and S4 Figs for controls with shuffled and shifted detected cycle times). The normalized cycle shapes show how much the spiking modulation during each cycle type deviates from the baseline (i.e. chance level). Notably, spiking modulation decreases progressively across timescales, with stronger modulation in faster rhythms like gamma and weaker modulation in slower ones like delta (Fig 1L).

Do the detected cycles correspond to the known rhythmic patterns in the LFP? In the average cycle shapes, the recurring peaks, with positive and negative delays are observed relative to the main peak (Fig 1D). This indicates that it is likely to observe one cycle type right after the occurrence of the same cycle type, suggesting that these cycles of activity are recurring events. To characterize their rhythmicity, we examined the distribution of inter-cycle intervals (ICI), which measures the delays between consecutive cycles. These distributions show an isolated peak near the expected cycle width. For example, the peak of the ICI distribution for the gamma cycles is around 20 ms, consistent with the width of the gamma cycle (Fig 1E). To further examine the relationship between the detected cycles in the spiking activity and the rhythmic patterns in the LFP, we calculated the average of LFP around the spiking-based detected cycles for each cycle type (Fig 1F). The average LFP around the

detected cycles follows the cyclic patterns that we observed in the spiking activity. In faster cycles, such as gamma and beta, temporal delays in the spiking-based-detected LFP rhythms across the cortical layers suggest a propagation from deeper layers to more superficial layers [27,28]. We also compared the patterns of Current Source Density (CSD) corresponding to the averaged LFPs around the cycles detected from the population spiking (Fig 1G). The distinct configurations of sources and sinks across different cycle types suggests that different mechanisms underlie different cycle types.

How are various cycle types interrelated? For example, does the presence of a theta cycle influence the likelihood of observing a gamma cycle? It has been widely reported that neural rhythms across different bands are interrelated [29–36]: different rhythms are likely to happen simultaneously (amplitude coupling), and the faster rhythms tend to occur in certain phases of the slower rhythms (phase coupling). To investigate the cycle coupling, we examined the distribution of faster cycles given the presence of slower cycles (e.g., the distribution of gamma cycles conditioned on the observation of a theta cycle). We observed that faster cycles tend to occur at the peak of slower cycles, which provides the maximum efficiency in modulating spiking activity (Fig 1H–1K, S5 Fig and S6 Figs).

## Sensory-driven gamma cycles capture the variation in signal transfer time between the retina and V1 across trials

To study neural dynamics in response to sensory stimulation, it is crucial to accurately align repeating sensory stimulation trials. During sensory stimulation experiments, we present the stimuli in several trials, and then, to compare the neural activity across trials, we align the trials based on the time of stimulus onset. However, this alignment method might not always be accurate, since the time taken for sensory information to reach the neural recording site can vary between trials. As gamma cycles have been shown to carry bottom-up sensory information [28,37,38], we used isolated sensory-driven gamma cycles to realign the trials.

After the alignment of trials based on the time of the first sensory-driven gamma cycles, we observed that the network response to sensory input goes through an increase followed by a decrease to baseline during 20-ms epochs. We estimate the network response to visual stimulation by examining changes in the number of recorded spikes, pooled from all the recorded neurons. Analyzing the network response across trials, relative to stimulus onset (Fig 2A and 2E), we observed that population spiking starts to increase and rises to a peak with a delay relative to stimulus onset (e.g., 50 ms is the peak time in the example recording session presented in Fig 2). The peak time in the trial-average response is the average signal transfer time between the stimulus onset and the arrival of the stimulus-related sensory information to the recording site. We used the time of the gamma cycle nearest to the average delay as the time of the first sensory-driven gamma cycle in each trial. After realigning the trials based on the timing of the first sensory-driven gamma cycle, we observe that the network spiking undergoes a cyclic pattern, an increase followed by a return to baseline, in response to sensory stimulation (Fig 2B, 2C, and 2G). After correcting for the jitter across the trials based on the first gamma cycle (Fig 2C), we reorder the trials based on the timing of the next gamma cycle and the second band of activity appears (Fig 2D). The width of these cycles is close to the gamma, suggesting that the visual cortex receives sensory information in gamma-width packets. Reordering trials based on the timing of population firing rate peaks after the stimulus onset also reveals gamma cycles of activity (S8 Fig). While using the peak timing of population spiking for reordering and jitter correction can partially capture gamma cycle dynamics, it does not accurately align trials. This is evident in S8B Fig, where the gamma cycle is less distinct compared to Fig 2B, and in S8G Fig, where the jitter-corrected response does

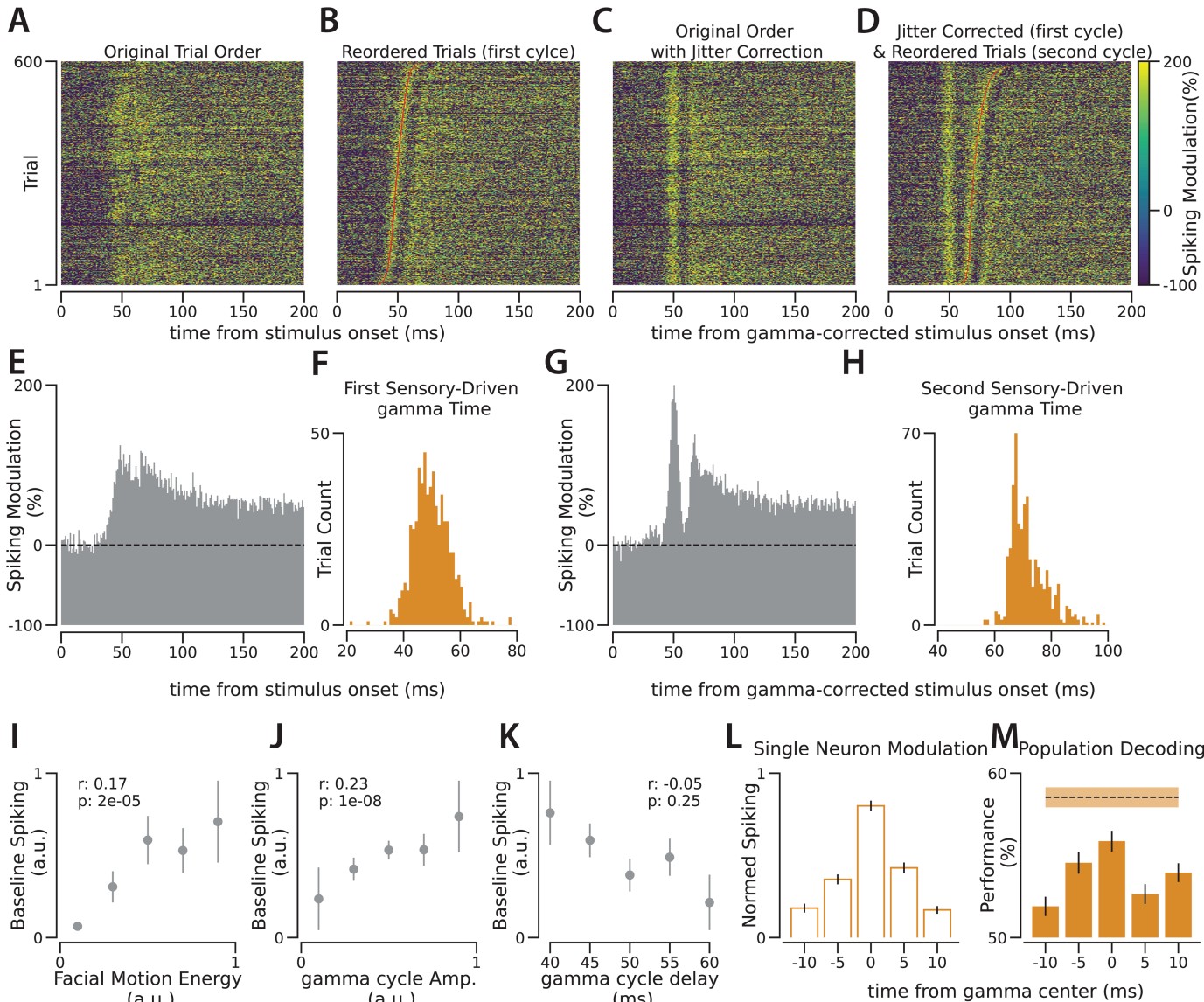

**Fig 2. Using sensory-driven gamma cycles to capture the biological jitter across trials. A.** Spiking response to visual stimulation where each 1-ms time bin depicts the combined spiking modulation from 131 simultaneously recorded V1 neurons. Each row represents a trial of visual stimulation, with spiking values relative to the 200-ms pre-stimulus window average. **B.** As in A, the trials are reordered according to the timing of the first sensory-driven gamma cycle. Red dots on each row indicate the center of these cycles. **C.** As in A, with the stimulus onset time adjusted based on varying sensory-driven gamma cycle timings across trials. **D.** As in C, the trials are reordered according to the timing of the second sensory-driven gamma cycle. Red dots on each row indicate the center of these cycles. **E.** Spiking average across trials, as shown in A. The y-axis shows the modulation of the population spiking relative to the baseline (represented by the dashed line). **F.** Distribution of the timing of the first sensory-driven gamma cycle across trials, relative to stimulus onset. **G.** As in E, but after adjusting stimulus onset based on varying sensory-driven gamma cycle timings across trials. **H.** Similar to F, distribution of the timing of the second sensory-driven gamma across trials, relative to the time of the stimulus onset corrected based on the time of the first gamma cycle in each trial. **I-K** Correlation of baseline spiking (from the 200-ms pre-stimulus window) across trials with **I.** facial motion, **J.** gamma cycle amplitude, and **K.** gamma cycle delay. Comparisons are made in 600 trials from the same recording session as other panels, and the correlation coefficients and the p-values result from Pearson correlation across these trials. The values of the baseline, facial motion, and gamma cycle amplitudes are normalized between 0 and 1. The trials are segmented into five groups within each panel; dots and lines represent the mean and s.e.m., respectively. **L.** Comparison of average spiking across 131 individual neurons for 5-ms bins during the first sensory-driven gamma cycle. The spiking of each neuron is normalized between 0 and 1 individually across the five time bins. **M.** Population decoding performance in determining stimulus orientation (horizontal vs. vertical) across 5-ms bins during the first sensory-driven gamma cycle (p-value of the t-test against the chance level is less than 0.0005 for all bins). The dashed line illustrates performance when spikes from the entire gamma cycle are combined. The shaded area and the lines represent s.e.m. For similar analysis from additional sessions, refer to S9 and S10 Figs.

not return to baseline. Additionally, this method is less effective at distinguishing consecutive cycles. Specifically, the second gamma bands detected from the peak of population spiking span 60 ms - three times the typical gamma duration (S8H Fig) - indicating that consecutive bands cannot be reliably separated by using the peak of population spiking. Cycle detection outperforms peak detection because consecutive cycles do not always manifest as distinct peaks.

To assess whether an individual sensory-driven gamma cycle carries sensory information, we performed population decoding in 5-ms bins during the first sensory-driven gamma cycle. In the recording sessions with drifting gratings as visual stimuli, we decoded horizontal versus vertical stimuli based on the activity of neurons in different phases of the gamma cycle. We found that the 5-ms central bin of the gamma cycle contains the most information about orientation, while the decoding capacity diminishes in both the preceding and subsequent phases of this 20-ms gamma cycle (Fig 2M). This suggests that sensory information is passed to the visual cortex in discrete packets [39,40].

Behavioral state has been shown to modulate sensory-evoked responses, as reflected in the locomotion [41], pupil size changes [42], and facial motion in mice [43]. Among these, facial motion is more temporally synchronized with network activity changes than pupil size [43] and is better at capturing subtle changes during non-locomotion periods. We observed that facial motion correlates with baseline population spiking rate across trials (Fig 2I) and it also correlates with the strength and timing of upcoming sensory-driven gamma cycles (Fig 2J–2K). Specifically, gamma cycles in high-arousal trials are stronger and arrive at the visual cortex faster.

## Correlated activity across brain regions in different time scales

Coherent neural activity, defined by synchronized rhythmic patterns, is widely regarded as a measure of information transfer between brain regions [44]. This coherence, typically measured using LFPs recorded in different brain regions, has been observed across various frequency bands and species [45–47] and has been linked to numerous cognitive functions [48–50]. In this study, we aimed to examine intra-areal coherence in the spiking activity across different time scales, corresponding to different frequency bands. To achieve this, we applied our cycle isolation method to a dataset featuring simultaneous recordings from multiple brain regions [26]. For each brain region, we identified cycles in the population spiking activity at different time scales (representing various frequency bands, see Fig 1A). Next, we calculated the average spiking patterns in other regions relative to the cycles detected in the reference region, analyzing each cycle type (corresponding to different frequency bands, see Fig 1A) separately.

Using the isolated cycles in the population spiking of a reference region, we constructed the distribution of spiking in other regions relative to each cycle type in the reference region. The generated distributions represent patterns of correlated spiking between pairs of regions, separately in each time scale. These intra-areal spiking correlation patterns reveal consistent and specific, time scale-dependent delays between the spiking of different regions. For example, as shown in Fig 3A, during the theta cycle, the spiking of the Anteromedial nucleus of the Thalamus (TH) is half a cycle out of phase relative to the spiking in the CA3 region of the Hippocampus (HPF). However, for the slower delta cycle, the spiking in these two regions is lagged by one-tenth of a cycle duration.

To quantify the delay between two regions for a specific cycle type (i.e. frequency band), we first checked whether the pattern of correlated spiking between the regions was significantly different from chance. If so, we estimated the delay as the time corresponding to the peak of

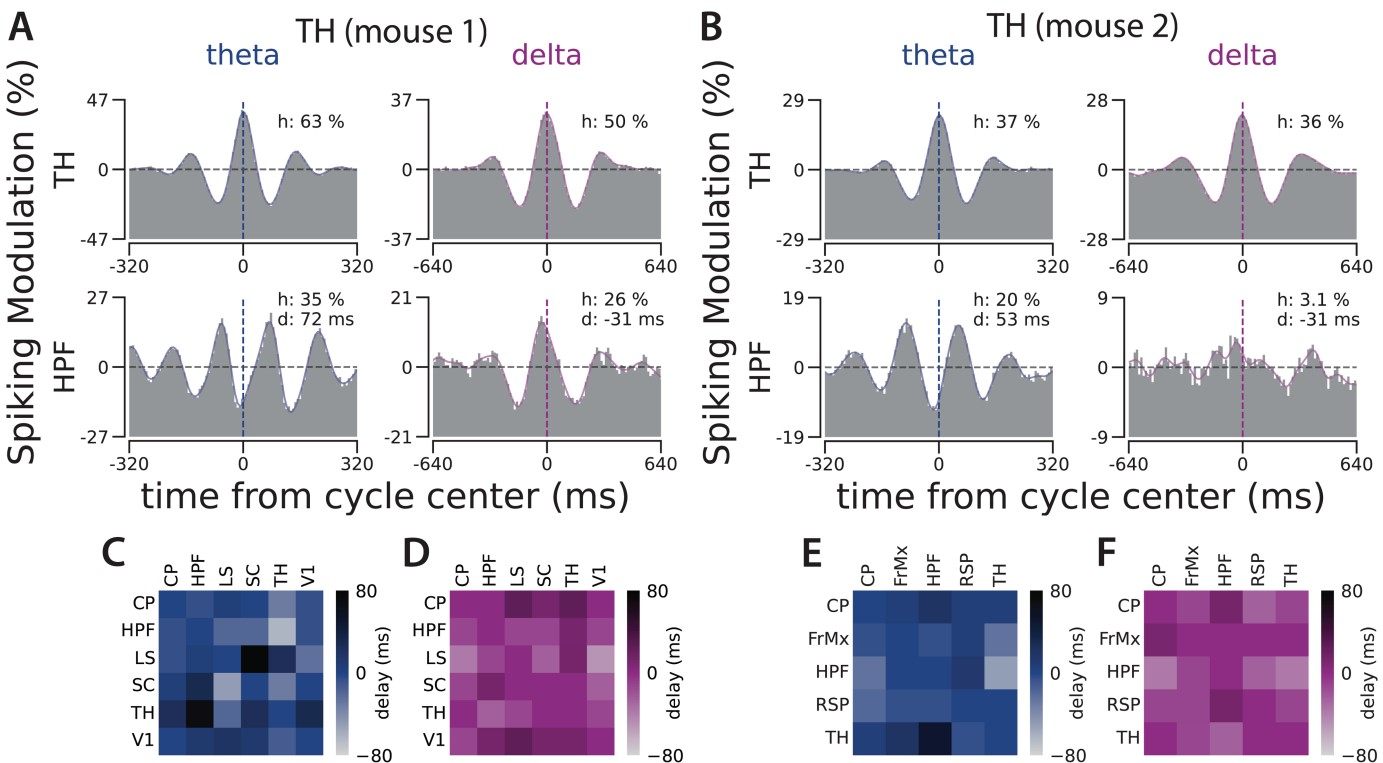

**Fig 3. Correlated dynamics between brain regions varies across different time scales. A.** The cycles detected in a recording site in the Anteromedial (AM) nucleus of the Thalamus (TH) were used to generate the distribution of spiking activity of two recording sites relative to theta and delta cycles in the TH: the first row is the distribution of spiking within the same Thalamical recording site relative to theta and delta cycles in the TH, the second row is the distribution of spiking in a recording site in the CA3 region of the Hippocampus (HPF) relative to the cycles in the TH. In each distribution, *h* represents trough-to-peak modulation relative to the chance level (the horizontal dashed line) and *d* represents the distribution peak delay relative to the center of the cycle in the reference region. **B.** Similar to A, but in a recording of a different animal, where cycles were detected at another AM Thalamic site. The distribution of spiking activities relative to these cycles is from the TH itself in the first row and the CA1 region of HPF in the second row. See S11 Fig for the results of the half-session control analysis for panels A and B. See S12, S13, and S14 Figs for the distribution of the spiking activity of all the recorded regions relative to the cycles detected at these recording sites in TH in these two animals and relative to a recording site in HPF in another animal. **C.** Pairwise delays between regions during theta cycles in the same recording as in A. **D.** Same as C for delays during the delta cycles. **E-F.** Same as C and D, for the same recording as in B. See S15 Fig for pairwise delays across all recorded regions for gamma to delta cycles in 3 animals.

the distribution of their correlated spiking, derived using the isolated cycles. To summarize the correlated spiking patterns across all region pairs, we constructed delay matrices for each cycle type (Fig 3C–3F show example delay matrices between specific regions in two animals for theta and delta bands, while S15 Fig presents the full delay matrices of all cycle types). For faster cycles like gamma and beta, coherent activity (albeit delayed) is evident only between recording sites in highly connected areas, such as visual areas in the cortex or different regions of the HPF. As we examine the slower cycles, we observe that an increasing number of regions exhibit coherence (S15 Fig).

## Isolating ultra slow cycles in population spiking, pupil size, and facial motion

By isolating cycles in neuronal spiking, we can also study slower rhythmic activities (< 0.1 Hz) typically filtered out in the LFP due to the limitations of recording hardware. Ultra-slow oscillations have been shown to be a significant feature of neural activity across brain

regions and species and correlate with cognitive functions and distinct network activity modes [19,51–56].

What are the cycle widths of the ultra-slow rhythms (Table 1)? It has long been noted that the bandwidth of neural rhythms is nearly doubled from one band to the next. Numbers between the Golden Ratio, 1.618 [57,58], and the Euler number, 2.7182 [8,59] have been suggested for this ratio between consecutive bands. The Golden Ratio, as the ratio between consecutive bands, seems to be an underestimation since it needs an introduction of subbands [58] that are not widely reported and the Euler number seems to be an overestimation since it misses the alpha-band [8]. So we decided to use two as a starting estimation point for the width of slow cycles that matches the experimental observations better and ensures the maximal efficacy of coupling across bands [60]. However, it should be noted that further work needs to be performed to verify the validity and uniqueness of these patterns of activity (see discussion).

As one moves from one band to the next slower one, the spectral bandwidth decreases by half, and the cycle duration of these rhythms doubles (e.g., around 20 ms for gamma cycles, 40 ms for beta cycles, and so on, Fig 1A–1B). Starting from a 20-ms cycle duration for gamma and doubling it, we arrive at a rhythm with a cycle duration of around 2.6 seconds, which is called "slow 2" by [8]. To name a broader range of slow rhythms, we termed it the *e* cycle (with reference to the Euler number) and named the other non-named slow cycles with reference to this one as *quarter e*, *half e*, 2*e*, 4*e*, 8*e*, 16*e*, and so on (Table 1).

The cycle detection method that we introduced for population spiking can also be applied to changes in pupil size and the facial movements of mice. Here, we present the results of the cycle isolation of the ultra-slow rhythms in population spiking of the recorded neurons in mouse V1 and S1. We also show results from searching for similar cycles in pupil size changes and facial motion (Fig 4 and S16, S17, S18, S19 Figs). These were recorded simultaneously with extracellular activity. The average activity around the detected cycles in each distinct ultra-slow time scale indicates a significant modulation in spiking activity, pupil size, and facial motion, although these modulations are less pronounced than in the faster cycles. The coherence across the modalities (spiking vs. pupil size vs. facial motion) during each cycle type indicates specific correlated dynamics. In particular, during ultra-slow cycles in both V1 and S1, the population spiking is in synchronized coherence with facial motion but is followed by the pupil with a delay of up to 1 second [43].

The ultra-slow cycles detected in spiking across the mouse brain reveal a specific order of activation of regions during multi-second cycles. We isolated the ultra-slow cycles (Table 1) individually in each brain area during spontaneous activity recordings with eight neuropixel probes [26]. We observed specific temporal dynamics of correlated activity across brain regions, which depend on the time scales. For instance, as shown in Fig 5, the two recording sites in TH and HPF are 600 ms delayed during the *e* cycles, but 3.8 s delayed during the 4*e* cycles. Such delays across regions might point to the propagation of the signal between

**Table 1.** Approximate cycle durations

| cycle name | duration | | cycle name | duration | | cycle name | duration | |
|---|---|---|---|---|---|---|---|---|
| *gamma* | 20 | ms | *quarter e* | 640 | ms | 8*e* | 20 | s |
| *beta* | 40 | ms | *half e* | 1.25 | s | 16*e* | 40 | s |
| *alpha* | 80 | ms | *e* | 2.5 | s | 32*e* | 80 | s |
| *theta* | 160 | ms | 2*e* | 5 | s | 64*e* | 160 | s |
| *delta* | 320 | ms | 4*e* | 10 | s | 128*e* | 320 | s |

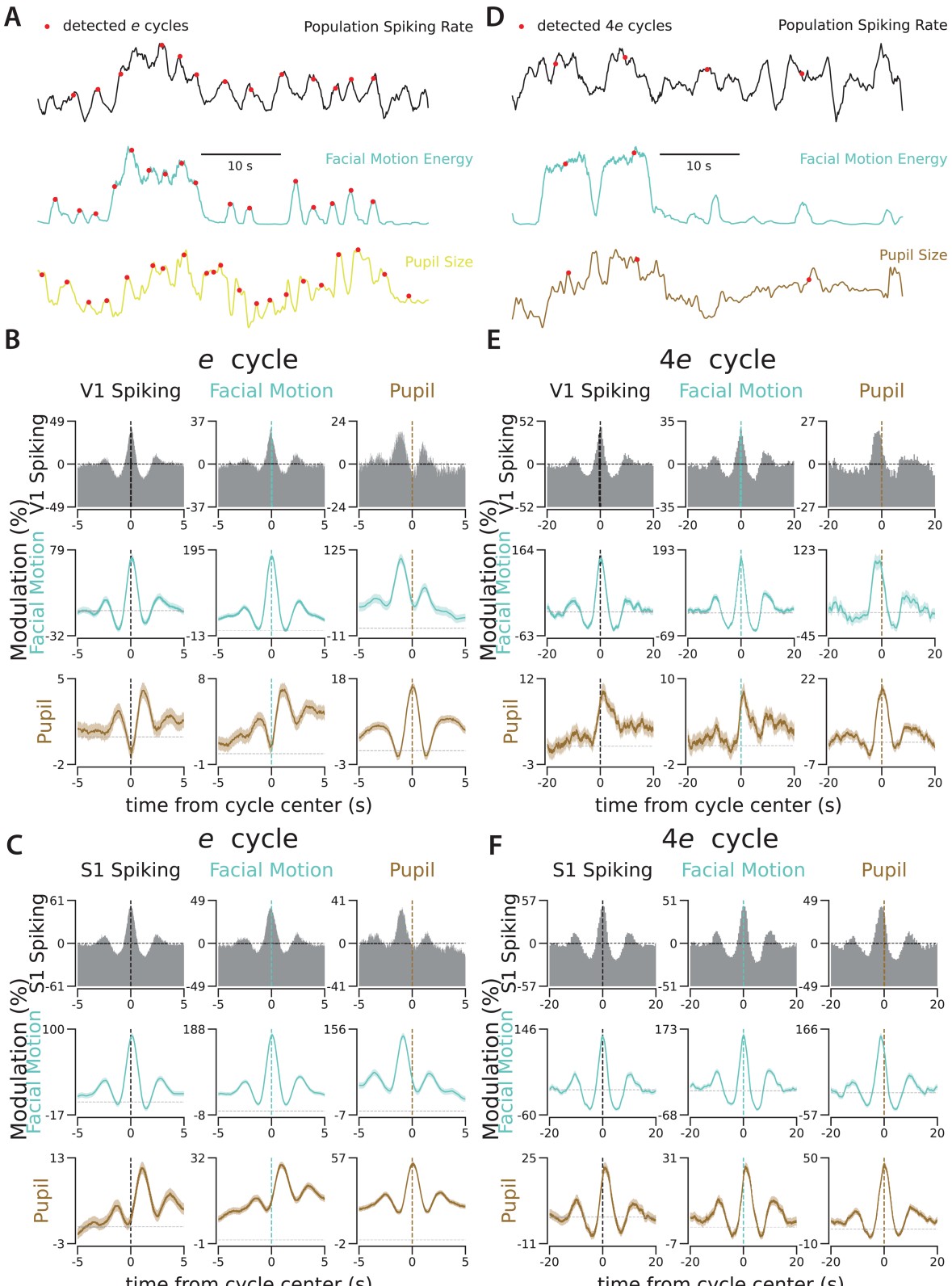

**Fig 4. Ultra-slow cycles in pupil size and facial motion appear with delays relative to similar cycles in V1 and S1. A.** Example traces of population spiking rate, facial motion, and pupil size changes in the sample recording from V1. The red dots indicate the centers of the *e* cycles detected individually on each trace. All traces are normalized between 0 and 1 in this example window. **B.** Each panel shows the

correlated activity between the target modality (e.g., facial motion signal on the second row) and the times of detected *e* cycles in the reference modality (e.g., V1 spiking in the first column). **C.** Same as B for a sample recording from S1. **D.** Same as A with red dots indicating the center of the detected 4*e* cycles. **E-F.** Same as B-C for 4*e* cycles. See S16, S17, S18, and S19 Figs for similar correlated activity during other slow cycles (Table 1) in these two recordings from V1 and S1. The vertical lines show the center of the detected cycles in the reference modality. The horizontal line in each panel shows the chance level. The vertical axes depict the modulation in the target modality relative to the chance level. The shaded area is s.e.m..

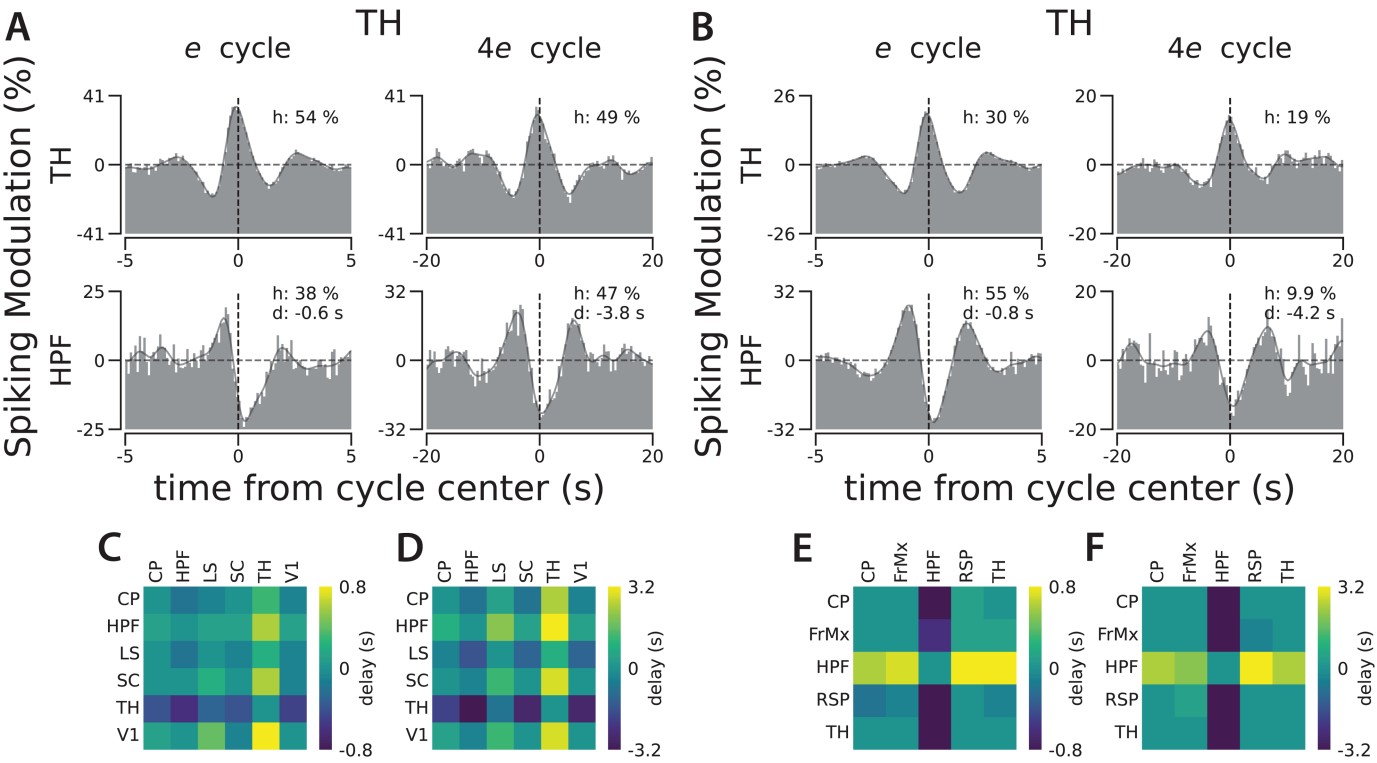

**Fig 5. Ultra-slow cycles are also delayed across regions. A.** Cycles detected in a recording site in the Thalamus (TH) were used to generate the spiking distribution in two recording sites relative to the *e* and 4*e* cycles in the TH: The first row is the distribution of spiking within the same Thalamic recording site relative to the *e* and 4*e* cycles in the TH, the second row is the distribution of spiking in a recording site in Hippocampus (HPF) relative to the cycles in the TH. In each distribution, *h* represents trough-to-peak modulation relative to the chance level (the horizontal dashed line) and *d* represents the distribution peak delay relative to the center of the cycle in the reference region. **B.** Similar to A, but in a recording of a different animal, cycles were detected at another Thalamic site. As in A, the first row depicts spiking from TH and the second from HPF. See S20, S21, S22 Figs for the distribution of spiking activity across all recorded regions relative to the *quarter e* to 4*e* cycles detected within the TH sites of these two animals and an HPF site from another. **C.** Pairwise delays across regions during *e* cycles, as observed in A's recording. **D.** Same as C for delays during 4*e* cycles. **E-F.** Same as C and D, as observed in B's recording. See S23 and S24 Figs for the pairwise delays across all recorded regions for the *quarter e* to 64*e* cycles in 3 animals.

regions or the inhibition of one region during the activation of the other region. We summarize the delays between all pairs of recorded regions in this dataset in delay matrices for each cycle type (S23 and S24 Figs). For additional profiles of correlated activity, refer to S20, S21, and S20 Figs. Profiles of correlated activity across all pairs of recorded regions, spanning all time scales (from gamma to 64*e*) and across 3 animals, can be found in [61].

## Discussion

We introduced a method for isolating single cycles of rhythmic patterns in the spiking of densely recorded populations of neurons. We applied this cycle isolation method to densely

recorded neurons across the width of the cortex in mice V1 and S1 using single-shank probes. We found that the cycles captured in the population spiking reflect similar rhythmic patterns observed in the LFP. One advantage of cycle detection in spiking activity, instead of LFP, is that the spatial distribution of contributing neurons can be specified; for example, it is possible to detect the cycles separately in superficial and deep cortical layers using the neurons in each layer, while the measured LFP in each layer is affected by the activity in other layers and cannot be used for independent cycle detection across layers. The isolated cycles also demonstrate the cross-frequency couplings by showing specific patterns of co-occurrence probability for pairs of different cycle types. In sensory stimulation experiments in V1, we used detected sensory-driven gamma cycles to realign the trials. This trial realignment compensates for the biological variation in the travel time of sensory signals between the retina and V1 and suggests that sensory information is conveyed in discrete temporal packets of activity. This idea is closely related to the proposal of temporal-packet-based activity [39,40], generalized across different time scales. We further applied this method to isolate multi-second cycles in population spiking, facial movements, and pupil size changes. Using another dataset [26], we also observed time-scale-dependent dynamics in correlated spiking across mouse brain regions. The brain-wide ultra-slow cycles, also reflected in pupil size and facial motion, may represent the physiological mechanisms driving behavioral states that are employed by neuromodulators such as Acetylcholine [62–66]. One future direction is to use spikes from functionally and/or anatomically distinct groups of neurons to identify the cyclic activity patterns. With this approach, we will be able to follow the propagation of the activity across the circuit.

Are neural rhythms essential features of neural systems that play a necessary role in information processing and transfer [67]? Or are they just epiphenomena of these processes, lacking a functional role [68]? Or do they fall somewhere in between? Neural rhythms and their correlation with brain function have primarily been identified through LFP and EEG measurements. However, spectral analyses and auto/cross correlograms of single neuron spiking have also revealed neural rhythms, particularly in networks with strong and persistent rhythmicity such as the theta rhythms of the rodent hippocampus [69] and the rhythms in the cat visual cortex [70]. Furthermore, with recent advancements in capturing single-trial dynamics in population activity [71,72], neural rhythms have been detected in single-cell spiking by correcting the across-trials jitters during cognitive tasks [73]. However, the debate about the possible causal role of neural rhythms in brain function continues [74].

To shed light on this controversy, it is helpful to consider a related question: How much of the rhythmicity observed in the LFP can be captured solely from spiking activity? The LFP reflects population-level activity, capturing rhythmicity when synchronized rhythmic input is delivered to the network by axons or when neighboring neurons share synchronized subthreshold oscillations. However, if the rhythmic signals are not synchronized, their contributions cancel out in the effective extracellular fields recorded by the LFP. In contrast, the spiking activity of individual neurons has limited power to detect such rhythmicity compared to the LFP. By aggregating the spikes from nearby cells, however, we can create a population measure of spiking activity. The population firing rate derived in this way, as proposed by [19,20], reflects shared variations in the membrane potentials of nearby neurons and offers a spiking-based measure of population activity. This measure can capture network cyclic activity in a manner comparable to the LFP. However, it is important to note that the relationship between cyclic activity in the LFP and neuronal spiking is not inherently intrinsic. Cases of dissociation between LFP oscillations and neuronal spiking have been observed. For example, beta-band activity in the primate motor cortex shows such dissociation [23,24].

Isolating individual cycles of neural rhythms in the time domain offers a deeper perspective on the dynamics of neural activity. Spectral analysis, the primary tool for detecting rhythmicity, is particularly effective when strong similar cycles occur multiple times right after each other. Spectral analysis successfully captures neural rhythms, especially when strong rhythms are locked to an external event and the noise has been reduced by averaging across trials. However, in single-trial spike trains, if similar cycles that modulate the probability of network spiking are irregular and of variable duration, the power spectrum may not capture the cycles as the underlying basis functions (S1A Fig). Thus, a robust time-domain method that detects single cycles in such conditions is desirable. Time domain methods developed to capture individual events in LFP and EEG [16,17] have been used to characterize neural cyclic events, such as their rate, shape, and contribution to the power spectrum [75–77].

We utilized an LSTM network, a powerful machine learning tool to incorporate the history of signal for event detection, in the time-domain to estimate the likelihood of a specific cycle type at each moment within a spike train. The main assumptions that have been introduced to our decoder during training are that 1) cycles have a Gaussian shape, and 2) there is a doubling relation between consecutive cycles. These assumptions are biologically plausible and match the experimental observations [60]. To further validate these assumptions, we next need to employ an unsupervised method to identify the optimal basis functions that can best reconstruct local population spiking and/or LFP. Once these basis functions are identified, we can then develop customized decoders for cycle detection, which might vary across species and brain regions. If these cyclic patterns of activity are proven to be the basis functions of the neural signal, it would suggest a biophysical constraint on the neural activity patterns: a population of neurons can just be activated by temporal patterns that are a linear summation of a specific set of cycles. Another assumption that requires further investigation is the absence of inverse cyclic activity in our model, which would represent the net inhibitory input to the network that reduces the probability of population spiking with a cyclic pattern.

In this work, we demonstrated the ability to extract and align cycles of neural activity directly from neuronal spiking in a local population. For cycle isolation, we used a time-domain approach with an LSTM network trained on synthesized data. This strategy has allowed us to: 1) compensate for jitter in signal transmission time from the retina to V1, 2) demonstrate the time-scale dependency of correlated activity dynamics across mouse brain regions, and 3) reveal the cyclic correlation between neural activity and behavioral measures of arousal in slow time scales.

## Methods

### Animals, headpost surgery, and treadmill habituation

All animal handling procedures were performed in accordance with guidelines approved by the Albert Einstein College of Medicine Institutional Animal Care and Use Committee (protocol 00001393) and federal guide. We handled wild-type male mice, aged 3 to 5 months, for 10 to 15 minutes daily for three days prior to headpost surgery. On the day of surgery, the mouse was anesthetized with isoflurane and the scalp was cleaned with Betadine solution. After making a midline incision, the scalp was resected to expose the skull. A tungsten wire (50 $\mu$m) was then inserted into the right cerebellum through a small hole. This tungsten wire was connected to a gold-plated pin (World Precision Instruments) to serve as the ground/reference connection during electrophysiology recording. The reference pin and a custom 3D printed headpost were affixed to the skull using dental cement. However, the target recording site (either V1 or S1 in the left hemisphere) was left free from cement and

was merely sealed with vetbond. After surgery, analgesics were administered to help with recovery. Following a 3- to 5-day recovery period after surgery, the mice began treadmill training. Over the next four days, we gradually increased the time the mice were head-fixed on the treadmill, continuing until they appeared comfortable and started running without obvious stress.

## Electrophysiology, behavioral state monitoring and visual stimulation

Approximately 16-20 hours before the recording, a small craniotomy was performed over the designated recording site under light anesthesia using isoflurane, ensuring the dura remained intact. After craniotomy, the site was sealed with Kwik-Cast (World Precision Instruments), and the animal was allowed to recover in preparation for the recording session the next day. Single-shank 64-channel silicon probes (Sharpened H3, Cambridge NeuroTech) were used for the recordings. The probe was inserted 1200 $\mu$m deep from the dura at a rate of 1 $\mu$m/s and then retracted by 100 $\mu$m for faster stabilization. The broadband signal was recorded using an RHD USB interface board (Intan) at a rate of 20 kSample/s. For the LFP, the broadband signal from each channel was low-pass filtered (< 200 Hz) and downsampled to 2 kSample/s. To reduce shared high-frequency noise, the signal from each channel was high-pass filtered (> 300 Hz) and subtracted by the median of the high-pass filtered signals across the 64 channels at each time point. These median-corrected high-pass filtered signals were then processed to extract spike times using KiloSort [78]. The KiloSort output was manually curated to discard non-spike patterns.

The left half of the mouse face was recorded with the BFLY-PGE-13S2M-CS camera (FLIR) at 30 frames/s under infrared illumination. The recorded frames were synchronized with the electrophysiology signal using the TTL pulses the camera sends out during each frame's lens opening. The facemap software was used to extract the pupil size and facial motion [79]. To track facial movement, a rectangular window containing the whiskers and nose was selected.

For visual stimulation, a 1080x1920p gamma-corrected portrait mode Asus ROG PG248Q monitor was placed 20 cm from the right eye of the animal. This monitor was displayed at 180 Hz (inter-frame-interval < 6 ms) with a GTX 1080Ti graphic card with g-sync disabled. Custom Matlab code with Psychtoolbox [80] was used for stimulus presentation. The stimulus onset was detected using an SM1PD1A photodiode (Thorlabs). The rise time of the stimulus onset was measured with the photodiode signal being less than 2 ms. For visual stimulation, drifting gratings or natural images were presented to the animal. In the drifting grating sessions, 12 directions (30° apart starting from 0) were used. The drifting gratings consisted of equal-sized (5 cm wide) full-contrast white and black bars with a temporal frequency of 2 Hz. Each direction was presented for 2 seconds in 50 trials with a variable 1.5-2 seconds gray screen intertrial interval. The order of 600 trials was randomized. In sessions with natural stimuli, 70 stimuli from ImageNet [81] were presented to the animal. Each stimulus was presented for 1 second in 27 trials with a variable 0.75-1 second gray screen inter-trial interval. The order of 1890 trials was randomized.

To evaluate neural activity in response to visual stimulation, the change in spiking after stimulus onset was quantified relative to baseline spiking. The baseline was defined as the average spiking in the 200 ms window prior to stimulus onset. To estimate the network response (Fig 2A–2H), spikes from all neurons recorded in each session were pooled together. For the population decoding of the orientations presented in 5-ms bins (Fig 2M), the 12 directions were reduced to 2 orientations (horizontal vs. vertical) by combining nearby orientations. This reduction was necessary because the number of spikes from each neuron in

5-ms bins is very sparse to distinguish between many conditions when the number of recorded neurons (55 neurons on average in our dataset) is limited. A Support Vector Machine (SVM) was then used to decode the presented orientation from the neuronal spiking [82]. The training-to-testing trial ratio was set at 4:1 and the train/test was repeated 100 times, randomly selecting trials in each iteration.

## Cycle detection

For the cycle isolation, a network of Long Short Term Memory (LSTM) units was trained on synthetic data using the TensorFlow package [83,84]. The LSTM network consisted of two layers of 200 units each. The input is sampled at 30 samples/(target cycle). The input to the network at each time point consists of 61 samples, including 30 samples before and 30 samples after the current time point. The output layer of the network consists of one dense unit (one sample corresponding to the center sample of the window provided as the input at each step) with linear activation that reflects the probability of the detection of the target cycle at each time point. 85% of the synthetic data was used for the training and 15% was used for testing until the convergence based on the accuracy. The batch size was set to 128. Root Mean Square Propagation (RMSprop) was used for the optimization and Mean Square Error (MSE) was used as the loss function. To introduce noise to the signal during training, a Gaussian Noise layer (stddev: 0.05) was added immediately after the input layer. A synthesized signal with 3 million samples was used to train the network. The synthesized signal was generated using seven cycle types as basis functions. The duration of each cycle type doubled relative to the previous one (3 cycles shorter and three cycles longer than the target cycle for detection). Each type of cycle occurred with a 60% chance. Different cycle types are randomly combined in all possible ways, while cycles of the same type do not overlap. The width of each cycle at each instance was randomly jittered by adding a value derived from a normal distribution with $\mu = 0$ and $\sigma$ set to 30% of the cycle width. These parameters were chosen so that the spectral density of the synthesized signal matches the average spectral density of the population firing rate at 29 recording sites across the mice brain from two datasets (S1A Fig):

$$s_{\text{train}}(t) = \text{Normalize}\left(\sum_{i=1}^{7}\sum_{j=1}^{N_i} \alpha_{ij} c_{ij}(t - t_{ij})\right) + \eta(t)$$

where:

- $c_{ij}(t)$: The Gaussian function representing the $j$-th instance of the $i$-th cycle type, defined as:

$$c_{ij}(t) = \exp\left(-\frac{1}{2}\left(\frac{t}{\sigma_{ij}}\right)^2\right)$$

- $\sigma_{ij}$: The jittered standard deviation (width) for the $j$-th instance of the $i$-th cycle type, defined as:

$$\sigma_{ij} = \sigma_i + \Delta\sigma_{ij}, \quad \Delta\sigma_{ij} \sim \mathcal{N}(0, 0.3 \cdot \sigma_i)$$

    where $\sigma_i = \frac{d_i}{5}$.
- $d_i$: The base duration of the $i$-th cycle type, given by:

$$d_i = 2^{i-4} \cdot d_4$$

Here, $d_4$ is the duration of the target cycle.

- $d_{ij}$: The jittered duration of the $j$-th instance of the $i$-th cycle type, related to $\sigma_{ij}$ as:

$$d_{ij} = 5 \cdot \sigma_{ij}$$

- $t_{ij}$: The center time of the $j$-th instance of the $i$-th cycle type, ensuring that instances of similar cycle types don't overlap:

$$t_{ij} = t_{i(j-1)} + d_{i(j-1)}$$

$t_{4j}$ time points are the targets of decoding.

- $\alpha_{ij}$: A binary inclusion variable sampled as:

$$\alpha_{ij} \sim \text{Bernoulli}(p), \quad p = 0.6$$

- Normalize($\cdot$): The normalization function scales the synthesized signal to lie between 0 and 1:

$$\text{Normalize}(s) = \frac{s - \min(s)}{\max(s) - \min(s)}$$

- $\eta(t)$: Additive Gaussian noise sampled as:

$$\eta(t) \sim \mathcal{N}(0, \sigma^2), \quad \sigma = 0.05$$

For training, the network output was set as a binary one-dimensional time series of the same length as the input. This output signal was set to zero everywhere except at the center of the target cycle.

$$y(t) = \begin{cases} 1, & \text{if } t = t_{4j} \text{ (center of the target cycle)} \\ 0, & \text{otherwise} \end{cases}$$

The sampling rate of the synthesized signal was set so that the target cycle was sampled 30 times per cycle. For predicting each output sample 61 samples from the input were provided to the network including 30 samples before and 30 samples after the target sample. A Mean Squared Error (MSE) loss function was used for the training:

$$\mathcal{L} = \frac{1}{T} \sum_{t=1}^{T} \left( y(t) - \hat{y}(t) \right)^2$$

where:

- $\hat{y}(t)$: The predicted output of the decoder at time $t$.

After training, network performance was evaluated using a ground-truth signal. An optimal threshold was chosen, based on the receiver operating characteristic (ROC) curve of the decoder, to decide the detected cycle times of the network output (S1B–S1D Fig). The trained network, along with the functions to apply it to neuronal spiking, is accessible at github.com/esiabri/isoCycle.

To quantify the average shape of each cycle (defined as the population spiking distribution relative to detected cycle times and normalized by chance level), we measured the height change from the left trough to the peak of the distribution (Fig 1D). Similarly, to quantify the coherence between two regions (defined as the population spiking distribution in the target region relative to the detected cycle times in the reference region and normalized to the chance level), we measured the trough-to-peak height and the delay of the peak of the distribution (see the bottom rows in Fig 3A–3B for example coherence patterns between recording sites in TH and HPF). The coherence patterns, as well as the quantified coherence strength and delay for all region pairs and cycles (from gamma to 64$e$) of the three animals in [26], are available in [61]. The quantification of the coherence pattern was performed only if it exceeded a p-value threshold of <0.01 in the Kolmogorov-Smirnov test compared to a uniform distribution. In each coherence pattern, the insets provide details on the exact p-value ($p$), coherence strength ($h$), and delay ($d$) (for examples, see S12, S13, S14, S20, S21, S22 Figs).

## Supporting information

**S1 Fig. Decoder Characteristics. A.** The blue trace represents the average power spectrum of the population spiking rate from 29 recording sites (4 S1 recording sites, 5 V1 recording sites, and 22 recording sites across the brain in Mouse 1 from [26]). The black trace is the power spectrum of the generated ground-truth. The ground truth was generated using six cycle types (high gamma, gamma, beta, alpha, theta, and delta see Fig 1A). Each cycle type occurred with a 60% chance. The width of each cycle at each instance was randomly jittered by adding a value derived from a normal distribution with $\mu = 0$ and $\sigma$ equal to 30% of the cycle width (e.g. 6.6 ms for the gamma cycle). After normalizing the signal to range between 0 and 1, the noise was introduced by adding a random value, derived from a normal distribution with $\mu = 0$ and $\sigma = 0.05$, at each time point. **B.** An example period of the ground truth spiking probability. The decoder was run to detect the gamma cycles on this signal. The orange dots show the center of each implemented gamma cycle in the signal, and the red dots are the detected gamma cycles after thresholding and peak detection on the output of the decoder (see the next panels). **C.** The decoder's ROC curve with the threshold used for the cycle detection. **D.** The decoder output (red trace) in response to the signal in B is compared to the ideal output (orange trace), which is zero everywhere except at the center of the gamma cycles. The dashed line is the threshold level (panel C) chosen to decide the cycle time based on the decoder output.
(TIFF)

**S2 Fig. Comparison with Other Methods. A.** An example period of the ground truth spiking probability (same as S1B Fig). The *bycyle* method [16] was run to detect the gamma cycles on this signal. The orange dots show the center of each implemented gamma cycle in the signal, and the red dots are the detected gamma cycles. The implemented gamma cycles were on average 20 ms in duration. The original signal was low pass filtered, using *neurodsp.filt.filter_signal* with *f_lowpass=150, n_seconds=0.1* and then was analyzed with *bycycle.cyclepoints.find_extrema* using the following parameters: *fs=1280, f_range=(40,60), filter_kwargs='n_seconds':0.1*. **B.** The average ROC curve was calculated by using 10 synthetic signals. *bycycle* doesn't provide different outputs based on varying thresholds, so the range of the ROC curve for the *bycycle* is limited. **C.** Similar to A, red dots indicate the signal's peaks when bandpass filtered between 40 and 60 Hz. **D.** Similar to B for comparison of the ROC

curve between our method and bandpass filtering while using varying thresholds to identify the signals' peaks. The partial Area Under the Curve (pAUC) of the ROC for both methods was calculated by normalizing to each method's maximum False Alarm rate, as their ROC curves are bounded. isoCycle: $0.975 \pm 0.002$ and band pass filtering: $0.879 \pm 0.005$. Shaded areas and error bars show standard deviations.
(TIFF)

**S3 Fig. Spiking Modulations Disappear with Shifted Cycle Times.** Same analyses as in panels D–F of Fig 1, when a random jitter of up to half a cycle width is added to or subtracted from each detected cycle time.
(TIFF)

**S4 Fig. Spiking Modulations Disappear with Shuffled Cycle Times.** Same analyses as in panels **D–F** of Fig 1, when the detected cycle times were shuffled while the distribution of the inter-cycle intervals has been kept the same as for the original cycle times.
(TIFF)

**S5 Fig. Relations between all Pairs of Cycles. A.** Similar to Fig 1G, between all pairs of cycles (p = 0.5 used as the threshold level for cycle detection in this figure to reduce the chance of false alarms).
(TIFF)

**S6 Fig. Distributions of Spiking Around the Times of Co-occurrence of the Cycle Pairs as in S5 Fig.** Similar to Fig 1H, with times of co-occurrence for all pairs of cycles (p = 0.1 used as the threshold level for cycle detection in this figure).
(TIFF)

**S7 Fig. Cycle Detection in Primary Somatosensory Cortex (S1).** Similar to Fig 1D–1E, for cycles detected from 49 neurons recorded simultaneously in mouse S1 for 8370 seconds.
(TIFF)

**S8 Fig. Reordering trials based on the peak in the population firing rate A.** Spiking response to visual stimulation where each 1-ms time bin depicts the combined spiking modulation from 131 simultaneously recorded V1 neurons. Each row represents a trial of visual stimulation, with spiking values relative to the 200-ms pre-stimulus window average. **B.** As in A, the trials are reordered according to the timing of the peak of the population firing rate in the 20 to 60 ms window after the stimulus onset in each trial. Red dots on each row indicate the time of the peaks in each trial. **C.** As in A, with the stimulus onset time adjusted based on varying peak timings across trials. **D.** As in C, the trials are reordered according to the timing of the peak of the population firing rate in the 60 and 100 ms window after the stimulus onset. Red dots on each row indicate the time of the peaks. **E.** Spiking average across trials, as shown in A. The y-axis shows the modulation of the population spiking relative to the baseline (represented by the dashed line). **F.** Distribution of the timing of the first sensory-driven peak across trials, relative to stimulus onset. **G.** As in E, but after adjusting stimulus onset based on varying peak timings across trials. **H.** Distribution of the timing of the second sensory-driven peak across trials, relative to stimulus onset.
(TIFF)

**S9 Fig. Extra Experiment for Fig 2.** Similar analyses as in Fig 2 for a recording in another animal. In this recording session, 58 neurons were simultaneously recorded while drifting gratings were shown to the animal.
(TIFF)

**S10 Fig. Extra Experiment for Fig 2**. Similar analyses as in Fig 2 for a recording in another animal. In this recording session, 56 neurons have been simultaneously recorded, while 70 distinct natural stimuli during 1890 trials have been shown to the animal. (TIFF)vspace*6pt

**S11 Fig. Half Session Controls for Fig 3A–3B.** Similar to Fig 3A–3B when the cycles detected in each session's first and second half have been used separately to compute the spiking distribution. **A–B.** First half. **C–D.** Second half.
(TIFF)

**S12 Fig. All Regions Relative to The Cycles in Thalamus.** Similar to Fig 3A for all the recorded regions relative to the cycles detected in Thalamaus (site TH_3 in mouse 1, named Kerbs, in the dataset from [26]). If there is more than one recording site in one brain region, the recording sites are differentiated with a number added to the end of the region name (e.g., TH_1, TH_2, etc.). Similar figures for all regions in this animal can be found at [61].
(TIFF)

**S13 Fig. All Regions Relative to The Cycles in Thalamus.** Similar to Fig 3B for all the recorded regions relative to the cycles detected in Thalamus (site TH_1 in mouse 3, named Robbins, in the dataset from [26]). If there is more than one recording site in one brain region, the recording sites are differentiated with a number added to the end of the region name (e.g., TH_1, TH_2, etc.). Similar figures for all regions in 3 animals can be found at [61].
(TIFF)

**S14 Fig. All Regions Relative to The Cycles in Hippocampus.** Similar to Fig 3A–3B for all the recorded regions relative to the cycles detected in Hippocampus (site HPF_2 in mouse 1, named Kerbs, in the dataset from [26]). If there is more than one recording site in one brain region, the recording sites are differentiated with a number added to the end of the region name (e.g., TH_1, TH_2, etc.). Similar figures for all regions in 3 animals can be found at [61].
(TIFF)

**S15 Fig. Across Regions Delay from gamma to delta for 3 Animals in [26].** Similar to Fig 3C for gamma, beta, alpha, theta, and delta cycles across all regions in 3 animals. If the pattern of correlated activity between two regions is not significant (see Methods), then the corresponding cell in the delay matrix is left blank. Patterns of correlated spiking across regions, from which these delays were derived, can be found at [61].
(TIFF)

**S16 Fig. Correlated Activity between Pupil Size, Facial Motion, and V1 Spiking During** *quarter e*, *half e*, **2*e*, and 8*e* Cycles.** Similar to Fig 4B & 4E for four other slow cycles.
(TIFF)

**S17 Fig. Correlated Activity between Pupil Size, Facial Motion, and V1 Spiking During 16*e*, 32*e*, 64*e*, and 128*e* Cycles.** Similar to Fig 4B & 4E for four other slow cycles.
(TIFF)

**S18 Fig. Correlated Activity between Pupil Size, Facial Motion, and S1 Spiking During** *quarter e*, *half e*, **2*e*, and 8*e* Cycles.** Similar to Fig 4C & 4F for four other slow cycles.
(TIFF)

**S19 Fig. Correlated Activity between Pupil Size, Facial Motion, and S1 Spiking During 16*e*, 32*e*, 64*e*, and 128*e* Cycles.** Similar to Fig 4C & 4F for four other slow cycles.
(TIFF)

**S20 Fig. All Regions Relative to The Cycles in Thalamus.** Similar to Fig 5A for all the recorded regions relative to the cycles detected in Thalamus (site TH_3 in mouse 1, named Kerbs, in the dataset from [26]). If there is more than one recording site in one brain region, the recording sites are differentiated with a number added to the end of the region name (e.g., TH_1, TH_2, etc.). Similar figures for all regions in 3 animals can be found at [61].
(TIFF)

**S21 Fig. All Regions Relative to The Cycles in Thalamus.** Similar to Fig 5B for all the recorded regions relative to the cycles detected in Thalamus (site TH_1 in mouse 3, named Robbins, in the dataset from [26]). If there is more than one recording site in one brain region, the recording sites are differentiated with a number added to the end of the region name (e.g., TH_1, TH_2, etc.). Similar figures for all regions in 3 animals can be found at [61].
(TIFF)

**S22 Fig. All Regions Relative to The Cycles in Hippocampus.** Similar to Fig 5A-B for all the recorded regions relative to the cycles detected in Hippocampus (site HPF_2 in mouse 2, named Waksman, in the dataset from [26]). If there is more than one recording site in one brain region, the recording sites are differentiated with a number added to the end of the region name (e.g., TH_1, TH_2, etc.). Similar figures for all regions in 3 animals can be found at [61].
(TIFF)

**S23 Fig. Across Regions Delay from *quartere* to 4*e* for 3 Animals in [26].** Similar to Fig 5C for *quartere*, *half e*, *e*, 2*e*, and 4*e* cycles across all regions in 3 animals. If the pattern of correlated activity between two regions is not significant (see Methods), then the corresponding cell in the delay matrix is left blank. Patterns of correlated activity between regions, from which these delays were derived, can be found at [61].
(TIFF)

**S24 Fig. Across Regions Delay from 8*e* to 128*e* for 3 Animals in [26].** Similar to Fig 5C for 8*e*, 16*e*, 32*e*, and 64*e* cycles across all regions in 3 animals. If the pattern of correlated activity between two regions is not significant (see Methods), then the corresponding cell in the delay matrix is left blank. Patterns of correlated activity between regions, from which these delays were derived, can be found at [61].
(TIFF)

## Supplementary Notes

**S1 Note. Cycles and Power Spectrum Schematics in Fig 1**.
(PDF)

## Author contributions

**Conceptualization:** Ehsan Sabri, Renata Batista-Brito.

**Data curation:** Ehsan Sabri.

**Formal analysis:** Ehsan Sabri.

**Funding acquisition:** Renata Batista-Brito.

**Investigation:** Ehsan Sabri.

**Methodology:** Ehsan Sabri.

**Project administration:** Renata Batista-Brito.

**Resources:** Renata Batista-Brito.

**Software:** Ehsan Sabri.

**Supervision:** Renata Batista-Brito.

**Validation:** Ehsan Sabri.

**Visualization:** Ehsan Sabri.

**Writing – original draft:** Ehsan Sabri, Renata Batista-Brito.

**Writing – review & editing:** Ehsan Sabri, Renata Batista-Brito.

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
