## [Decision Letter · Decision Letter 0]

25 Nov 2024

PCOMPBIOL-D-24-01889Isolating Single Cycles of Neural Oscillations in Population SpikingPLOS Computational BiologyDear Dr. Sabri, Thank you for submitting your manuscript to PLOS Computational Biology. After careful consideration, we feel that it has merit but does not fully meet PLOS Computational Biology's publication criteria as it currently stands. Therefore, we invite you to submit a revised version of the manuscript that addresses all the points raised during the review process. Please note that the reviewers are particularly concerned that the presented method covers a set of questions for which pretty standard signal processing methods are available. The revised manuscript must provide clear demonstrations that the new method outperforms existing signal processing methods in simulated case of clear neuroscientific relevance.  Please submit your revised manuscript within 60 days Jan 25 2025 11:59PM. If you will need more time than this to complete your revisions, please reply to this message or contact the journal office at ploscompbiol@plos.org. Please include the following items when submitting your revised manuscript: * A rebuttal letter that responds to each point raised by the editor and reviewer(s). You should upload this letter as a separate file labeled 'Response to Reviewers'. This file does not need to include responses to formatting updates and technical items listed in the 'Journal Requirements' section below. * A marked-up copy of your manuscript that highlights changes made to the original version. You should upload this as a separate file labeled 'Revised Manuscript with Track Changes'. * An unmarked version of your revised paper without tracked changes. You should upload this as a separate file labeled 'Manuscript'. If you would like to make changes to your financial disclosure, competing interests statement, or data availability statement, please make these updates within the submission form at the time of resubmission. Guidelines for resubmitting your figure files are available below the reviewer comments at the end of this letter. We look forward to receiving your revised manuscript. Kind regards,Stefano PanzeriAcademic EditorPLOS Computational Biology Daniele MarinazzoSection EditorPLOS Computational Biology Feilim Mac GabhannEditor-in-ChiefPLOS Computational Biology Jason PapinEditor-in-ChiefPLOS Computational Biology **Journal Requirements:**

At this stage, the following Authors/Authors require contributions: Ehsan Sabri, and Renata Batista-Brito. Please ensure that the full contributions of each author are acknowledged in the "Add/Edit/Remove Authors" section of our submission form.

3) Please provide an Author Summary. This should appear in your manuscript between the Abstract (if applicable) and the Introduction, and should be 150u2013200 words long. The aim should be to make your findings accessible to a wide audience that includes both scientists and non-scientists. Sample summaries can be found on our website under Submission Guidelines:

5) We have noticed that you have uploaded Supporting Information files, but you have not included a list of legends. Please add a full list of legends for your Supporting Information files after the references list.

6) We notice that your supplementary Figures are included in the manuscript file. Please remove them and upload them with the file type 'Supporting Information'. Please ensure that each Supporting Information file has a legend listed in the manuscript after the references list.

7) Thank you for uploading your study's underlying data set. We notice that there is a  CC BY-NC 4.0 license on your data. We would encourage you to consider using a license that is no more restrictive than CC BY, in line with PLOS’ recommendation on licensing (http://journals.plos.org/plosone/s/licenses-and-copyright). For a list of recommended repositories and additional information on PLOS standards for data deposition, please see [https://journals.plos.org/ploscompbiol/s/recommended-repositories

8) Please amend your detailed Financial Disclosure statement. This is published with the article. It must therefore be completed in full sentences and contain the exact wording you wish to be published.

2) State what role the funders took in the study. If the funders had no role in your study, please state: "The funders had no role in study design, data collection and analysis, decision to publish, or preparation of the manuscript.".

9) Please ensure that the funders and grant numbers match between the Financial Disclosure field and the Funding Information tab in your submission form. Note that the funders must be provided in the same order in both places as well. Currently, the funder "Whitehall Award" is missing from the Funding Information tab and the order of the funders do not exactly match in the Financial Disclosure field and the Funding Information tab.

Please indicate by return email the full and correct funding information for your study and confirm the order in which funding contributions should appear. Please be sure to indicate whether the funders played any role in the study design, data collection and analysis, decision to publish, or preparation of the manuscript.

**Reviewers' comments:**Reviewer's Responses to Questions

**Comments to the Authors:**

**Please note that the review is uploaded as an attachment.**

Reviewer #1: This study presents a LSTM-based cycle detection method and applies it on detecting the gama cycle to align gama cycles with trials and on calculating cycle-wise correlations across multiple brain regions and across time scales. Overall, the method seems novel. But some major flaws should be addressed before being further considered for publishing.

The reviewer thinks the method must be solidly validated before going further to use it on analyzing neuronal oscillations.

1. Is the method of synthesizing training data and test data with ground truth valid enough to represent neuronal oscillations? In other words, how to guarantee the trained LSTM-based algorithm are generalizable enough to be applied on real neuronal signals?

2. How to determine the hyperparameters, e.g. the sliding window length, noise level, batch size, optimizer used for training, the data splitting protocol for validation. And how many cycles on average are included in a batch?

3. What's the cost function of the algorithm?

4. What is the LSTM-based method's benefit comparing with SOTA methods?

Other questions on the following analysis are

What is the purpose of the following analysis? To validate the algorithm against the known phenomena or to find some novel results with the new tool? If the former, the authors are suggested to add further comments on what are the similarities between the results this study finds and the previously known phenomena. If the latter, it is highly suggested to find a way to validate the generalizability of the algorithm on real-world signals.

Reviewer #2: (Review is also attached as a ".docx" file with headers/bolding to make things easier to read.)

Summary

The authors propose a neural network-based approach to detect single cycles of oscillatory activity from population spiking data. The use of an LSTM architecture pre-trained with synthetic data is noteworthy, and demonstrates a novel application of decoding algorithms for posing scientific questions. There also seems to be a clear relationship between “single cycles” detected in the spiking, and the LFP/physiologic signals aligned to these cycles. The authors also provide intriguing interpretations on the computational role of correlations between population spiking cycles and different brain regions/recording modalities.

The major shortcoming of this manuscript is that it is applying a new neural network-based method for something that has standard alternatives in signal processing (e.g. using a spectrogram to detect transient increases in spectral power). While there is some brief justification on why transient oscillation/cycles may be difficult to detect, this needs to be shown explicitly rather than assumed. Further, this method is being used to characterize neural signals in a unique manner (oscillations of sum total population spiking), and interpret the biological significance. If the goal of this manuscript is to argue for the use of this method, then there needs to be concrete comparisons to existing tools, and a commentary on their strengths and weaknesses. If the goal is to characterize the significance of oscillations in population spiking, the same advice applies, as it’s difficult to assess if this is a quirk of the tool (and the assumptions of the synthetic training data), or a fundamental and biologically significant signal.

It should be noted that the writing and figures are of high quality. However, the manuscript would greatly benefit from reworking the definitions/descriptions of the LSTM method to add clarity. Given the current state of the manuscript, I recommend to accept with major revisions.

Main Comments

In all sections of the manuscript, it needs to be made obvious that 1) “population spiking” means the sum of all neuron firing rates (this term is ambiguous with respect to it being a multivariate or univariate signal), and 2) the oscillations detected are specific to this population-level spiking, and is not directly applied to LFP features. Throughout the introduction I was left thinking that this method inputs spiking data to the LSTM, and outputs predictions on LFP oscillations. The discussion results do a better job of making this distinction in some places, but in many places these details have to be inferred as the language used leaves room for ambiguity. A very specific example is L55 in the introduction which reads as very vague, I encourage you to add explicit definitions for these terms.

While it is certainly true that spiking in the local recorded population can produce a measurable change in the LFP, the reverse is not necessarily true as (e.g. subthreshold oscillations). Even in the suprathreshold case, motor cortical beta oscillations are a great example where there is a measurable change in local spiking, but it is a global reduction that is not phase locked to the oscillation. This is a really important caveat to mention in the introduction/discussion, as the paper seems to establish an equivalency between these signals. References specific to beta oscillations are below:

Rule, M. E., Vargas-Irwin, C. E., Donoghue, J. P., & Truccolo, W. (2017). Dissociation between sustained single-neuron spiking and transient β-LFP oscillations in primate motor cortex. Journal of Neurophysiology, 117(4), 1524-1543.

Confais, J., Malfait, N., Brochier, T., Riehle, A., & Kilavik, B. E. (2020). Is there an intrinsic relationship between LFP beta oscillation amplitude and firing rate of individual neurons in macaque motor cortex?. Cerebral Cortex Communications, 1(1), tgaa017.

Spectral analysis of firing rate summed across every neuron is not very common in computational neuroscience, especially applied to neuropixel data. It would be highly beneficial to add references that use this technique, up/down states during sleep come to mind. Otherwise it will be important to outline precisely why you think such a signal is important as it relates to neural oscillations.

It is a shame that there isn’t analysis in layer-specific differences in population spiking, as there is a rich literature on the unique contributions of deep/superficial layers to slow/fast oscillations. I’ll leave it to your discretion if this is within the scope of your paper (see my comment in the summary on focussing on methods vs. biological significance). If the focus of the paper is the biological significance, then a layer-specific analysis would provide much more compelling predictions on the biophysical mechanism relating population spiking to LFP oscillations. I don’t think single unit spike sorting would be necessary as it comes with many pitfalls. However, as you reference in the discussion, a major advantage of this technique compared to LFP’s is isolating the spatial location of neurons being summed over (of course I will challenge this statement as current source density analysis precisely aims to spatially localize LFP contributions).

There is a fundamental signal processing question that should be addressed: is an LSTM the right tool to detect transient oscillations? You cite cycle by cycle analysis (https://github.com/bycycle-tools/bycycle, Cole and Voytek 2019), why not apply this technique to the population spike rate as well? Other similar techniques with associated neuroscience publications include “burst detection” (https://github.com/danclab/burst_detection) and “spectral event analysis” (https://github.com/jonescompneurolab/SpectralEvents), both of which characterize transients in the time-frequency spectrogram. The simplest approach would be tracking threshold crossings in the envelope of a band-passed signal. A direct comparison to at least one existing approach would be highly informative and really strengthen the results of this paper. I certainly support the use of deep learning in science, but when analytic alternatives are available these neural network based approaches need to be benchmarked to build trust in the method. I’d be much more convinced if the LSTM outperforms standard methods in terms of AUC when applied to your synthetic dataset.

The github repository attached does not seem to include code to generate the synthetic data, or code for training the neural network. While it is clear that a lot of effort went into making this a usable python package, this repository does not allow me to reproduce the results in this manuscript. Even if the training code is not in a clean state, it is important to include this information to properly validate the claims in the manuscript.

Abstract

It is not a given that oscillations in EEG, LFP, and spiking have an “intrinsic” relationship. They can certainly correlate, but it is not universally true that neural oscillations are detectable on all 3 modalities given the same underlying mechanism. I think you should rewrite this to assert that oscillations in population spike rates are present, likely to be biologically significant, and should be treated as distinct form of neural activity in its own right. You can even state that there is a fundamental gap in understanding 1) how spiking oscillations relate to field potential oscillations, 2) how spiking oscillations are correlated across brain regions, and 3) how spiking oscillations are correlated to physiological signals and behavior.

It is strange to state that spikes “estimate the network’s activity over time”. It is a small sample of the entire network. Do you mean this is an estimate of the global network activity?

Introduction

Numerous researchers have developed methods to characterize the relationship between spiking and field potentials. Spike-field coherence is the most apparent method that is absent from the introduction/discussion. It will be important to add a paragraph outlining the commonly used methodology for relating spikes to field potentials, and precisely how your approach is unique from these methods.

The motivation for identifying “single cycles” is a bit vague and could be better stated. As it currently reads, this could be referring to either 1) detection of oscillatory bursts/events, or 2) accurate phase estimation in continuous rhythms (or perhaps both?). There is a growing body of literature emphasizing the importance of characterizing oscillations as bursts as opposed to rhythms (cycle by cycle analysis being one example that you cite) but this discussion is absent in the manuscript. It would help greatly to clarify what framework you’re working from, and add a few words on why treating neural oscillations as bursts or steady oscillations is important from a biological perspective. The discussion broaches this topic in more detail, but it would be useful to reference this work with 1-2 sentences in the introduction.

Results

Section 2.1

To better link these single cycles to a neural mechanism, it’d be highly beneficial to include a current source density (CSD) plot either in the main results, or as a supplemental figure. You make reference in L106 to some sort of propagation across layers. With a CSD plot you’d have a much more mechanistic description of this phenomenon in terms of current sinks/sources in different layers.

Section 2.2

The results here are interesting, but I feel a direct comparison to a simpler technique is necessary. A very naive approach would be to apply a peak finder to the population spiking signal, and apply the jitter correction/reordering to any peaks above a certain threshold. If you compare these techniques side by side, does your LSTM outperform the peak finder in terms of some alignment quality metric?

Figures 2I-K are not described in the text, what is the motivation behind these analyses?

In L149 it would be useful to add references to support the “discrete packets” hypothesis. The references in the discussion are great so just add them here.

Section 2.3

This is a cool analysis, but terms like “cycle type” and “time scales” make it hard to understand what’s being shown. This could be mitigated by very explicitly defining what you mean by “cycle type” (i.e. different frequency bands that are prominent in neural activity).

I’m still not entirely sure if “different time scales” refers to different cycle types, or the delay between regions. Again explicit definitions would help a lot.

The delay-matrices in Figure 3C-F are not described in the text, either remove these panels or give them a complete description.

Section 2.4

L175-L177: High pass filtering of LFP’s is a signal processing choice that is explicitly used to eliminate physiological signals like breathing and motion artifacts. It’s misleading to present population spiking as a better readout of ultra-slow signals when you can simply change the filter of LFP’s. Do you have any reason to think they might be a better read-out of ultra-slow oscillations? Of course I’ll note my earlier concern that oscillations in population spiking are not necessarily correlated with the same frequencies of oscillation in LFP, a caveat worth denoting in the introduction.

L184: Is “the value two for the ratio…” a typo?

The ideas of relating oscillations to euler’s number and the golden mean are quite intriguing. It would be much more convincing to show that these cycle widths actually correspond to something tangible in the frequency spectrum of the data rather than treating it as an a priori assumption (perhaps a spectrogram could be used to at least show transient increases in these ultra slow bands?). Given the power spectrum shown in supplemental figure S1A, I’m a bit suspicious that you could pick any cycle width and get something that looks reasonable. You touch on this problem in L259 of the discussion section, but the logic is somewhat strange given that in the results you argue these ratios are important based off of well observed peaks in the power spectrum of neural signals, but in the text you state that many things are missed when you look for peaks in the power spectrum. It just seems like a shaky justification for choosing these values, and highlighting these limitations in the discussion (or briefly in the results) will be useful for interpreting the significance of these findings.

In the Figure 5 legend, it is highly unusual to cite a link to figshare (Sabri 2023) that is specific to this paper. This is exactly why a supplemental section is included. If you don’t think it’s important enough for the supplemental section it might not be worth mentioning.

Either here or in the discussion, it’d be worth relating these delays to a biological mechanism. One possibility is they represent true delays in the propagation of neural signals between brain regions. Another is that one region inhibits the other, and there isn’t actually a delay in the sense of signal transfer speed.

Discussion

L248-258 is the motivation that I wish I had read in the introduction and really helps frame the subsequent analyses in the results. I highly encourage you to move this paragraph. The paragraph above provides important context, but I believe it can be condensed to 1-2 sentences when moved to the introduction (i.e. are oscillations “exhaust fumes”).

A very apparent future direction is applying a similar analysis at the single neuron level, and/or grouping spike rates of neurons according to some heuristic (location, cell-type/spike shape, tuning, etc.).

In L271-L282, I definitely agree that detecting the optimal set of “neural basis functions” is a very important goal. As I’ve outlined above, I think it is important to show that existing tools are insufficient for this task (the AUC metric on the synthetic data would provide a clear result).

In terms of identifying neural basis functions, it may be worth referencing that biophysical modeling is a promising approach towards this goal. The whole argument around characterizing aperiodic activity from Gao and Voytek builds from the idea that AMPA and GABA post synaptic potentials have distinct time constants, and the shapes (basis functions) of these signals influence the 1/f slope even when there’s no true rhythmicity. In a similar vein, biophysical models that attempt to understand the underlying mechanism of neural oscillations may prove to be an indispensable tool for constructing these basis functions from “first principles”, rather than generative models with no biological motivation. Below are some references of tools aimed at tackling this problem:

Lindén, H., Hagen, E., Łęski, S., Norheim, E. S., Pettersen, K. H., & Einevoll, G. T. (2014). LFPy: a tool for biophysical simulation of extracellular potentials generated by detailed model neurons. Frontiers in neuroinformatics, 7, 41.

Borges, F. S., Moreira, J. V., Takarabe, L. M., Lytton, W. W., & Dura-Bernal, S. (2022). Large-scale biophysically detailed model of somatosensory thalamocortical circuits in NetPyNE. Frontiers in Neuroinformatics, 16, 884245.

Neymotin, S. A., Daniels, D. S., Caldwell, B., McDougal, R. A., Carnevale, N. T., Jas, M., ... & Jones, S. R. (2020). Human Neocortical Neurosolver (HNN), a new software tool for interpreting the cellular and network origin of human MEG/EEG data. Elife, 9, e51214.

Dai, K., Gratiy, S. L., Billeh, Y. N., Xu, R., Cai, B., Cain, N., ... & Arkhipov, A. (2020). Brain Modeling ToolKit: An open source software suite for multiscale modeling of brain circuits. PLOS Computational Biology, 16(11), e1008386.

Similar to the above comment, for estimating neural basis functions in a purely data-driven approach, there is already some work on this in the EEG space which is very relevant:

Díaz, J., Ando, H., Han, G., Malyshevskaya, O., Hayashi, X., Letelier, J. C., ... & Vogt, K. E. (2023). Recovering Arrhythmic EEG Transients from Their Stochastic Interference. arXiv preprint arXiv:2303.07683.

Dupré la Tour, T., Moreau, T., Jas, M., & Gramfort, A. (2018). Multivariate convolutional sparse coding for electromagnetic brain signals. Advances in Neural Information Processing Systems, 31.

Methods

Section 4.2

In L325, is “this monitor was derived at 180 Hz” correct? Or is this meant to say “this monitor was displayed at 180 Hz”

Section 4.3

One of the major contributions of this paper is a novel decoder-based analysis, but the details of this analysis are buried in a single paragraph in the methods. As highlighted above it was rather difficult to parse what was actually being used to train and evaluate the network. These issues can largely be mitigated by including typeset equations for 1) the synthetic data generation, and 2) the loss function used to train the LSTM. As it currently stands the text in the introduction/results/methods are all too vague to properly evaluate this method.

Minor Comments

In Figure 2B I would use red dots to denote the center of the cycle, orange is too similar to the heatmap colors

In Figure 2D, why not denote the center of the second cycle?

Figure 2H seems to be a mistake and should be removed. Reordering should have no impact on the PSTH, and clearly this is identical to 2G.

Figure 2I-K, the caption description is mislabeled. I believe it should state “across trials with I. facial motion, J. gamma cycle amplitude… etc.”

Figure 2M, asterisks should indicate if the decoding performance is significantly above chance

Figure 3A-B: it would be useful to label something similar to “TH (mouse 1)” and “TH (mouse 2)” in the actual figure

**Have the authors made all data and (if applicable) computational code underlying the findings in their manuscript fully available?**

Reviewer #1: Yes

Reviewer #2: **No: **Code necessary to generate synthetic data, and train the LSTM decoder described in the manuscript does not seem to be available

PLOS authors have the option to publish the peer review history of their article (what does this mean?). If published, this will include your full peer review and any attached files.

Reviewer #1: **Yes: **Chunzhi Yi

Reviewer #2: **Yes: **Nicholas Tolley

**Figure resubmission:** While revising your submission, please upload your figure files to the Preflight Analysis and Conversion Engine (PACE) digital diagnostic tool, https://pacev2.apexcovantage.com/. PACE helps ensure that figures meet PLOS requirements. To use PACE, you must first register as a user. Registration is free. Then, login and navigate to the UPLOAD tab, where you will find detailed instructions on how to use the tool. If you encounter any issues or have any questions when using PACE, please email PLOS at figures@plos.org. Please note that Supporting Information files do not need this step. If there are other versions of figure files still present in your submission file inventory at resubmission, please replace them with the PACE-processed versions.
---

## [Decision Letter · Decision Letter 1]

5 Mar 2025

PCOMPBIOL-D-24-01889R1

Isolating Single Cycles of Neural Oscillations in Population Spiking

PLOS Computational Biology

Dear Dr. Sabri,

Thank you for submitting your manuscript to PLOS Computational Biology. After careful consideration, we feel that it has merit but does not fully meet PLOS Computational Biology's publication criteria as it currently stands. Therefore, we invite you to submit a revised version of the manuscript that addresses the points raised during the review process.

Please submit your revised manuscript within 60 days May 05 2025 11:59PM. If you will need more time than this to complete your revisions, please reply to this message or contact the journal office at ploscompbiol@plos.org. Please include the following items when submitting your revised manuscript:

We look forward to receiving your revised manuscript.

Kind regards,

Stefano Panzeri

Academic Editor

PLOS Computational Biology

Daniele Marinazzo

Section Editor

PLOS Computational Biology

**Additional Editor Comments :**

Dear Authors, the reviewers appreciated your revision effort but still major unaddressed concerns remain. You have one more round to revise the paper and satisfactorily address all the concerns. As it would be not appropriate to overload with work the reviewers, we expect that all the concerns will be address in full in this last round. Additional rounds of re-review may not be possible in case of a still unsatisfactory or incomplete revision.

**Reviewers' comments:**

Reviewer's Responses to Questions

**Comments to the Authors:**

**Please note that the reviews are uploaded as attachments.**

Reviewer #1: Thanks for the point-by-point reply. I appreciate the efforts paid by the authors on revising the manuscript. Major concerns still remain. On validating the LSTM algorithm, the DL- driven models learn the distribution of input potentially in a manner not similar to how human understand the input. That is, no matter how biologically plausible the generated waves are, we cannot guarantee the learned parameters of LSTM can be generalizable on real data, unless validation can be performed on real data. The reviewer would suggest to show the performance of segmenting waves on real data. Otherwise, please modify the statement on the following parts by indicating that the findings enabled by the algorithm may be biased by the learned bias and the findings are not guaranteed to reflect the intrinsic characteristics within data. The findings may reflect the learning bias induced by generated data.

Reviewer #2: review uploaded as attachment

**Have the authors made all data and (if applicable) computational code underlying the findings in their manuscript fully available?**

Reviewer #1: Yes

Reviewer #2: Yes

PLOS authors have the option to publish the peer review history of their article (what does this mean?). If published, this will include your full peer review and any attached files.

Reviewer #1: **Yes: **Chunzhi Yi

Reviewer #2: **Yes: **Nicholas Tolley

**Figure resubmission:**
---

## [Decision Letter · Decision Letter 2]

22 Apr 2025

Dear Dr Sabri,

We are pleased to inform you that your manuscript 'Isolating Single Cycles of Neural Oscillations in Population Spiking' has been provisionally accepted for publication in PLOS Computational Biology.

Best regards,

Stefano Panzeri

Academic Editor

PLOS Computational Biology

Daniele Marinazzo

Section Editor

PLOS Computational Biology

Reviewer's Responses to Questions

**Comments to the Authors:**

Reviewer #1: The authors have addressed all my concerns. Thanks for the efforts paid by the authors.

Reviewer #2: The authors have fully addressed all of my suggestions, my recommendation is to accept. I’ll be very excited to see this in print, congratulations!

There are some very minor mistakes I caught that can be corrected for the final proof, no need for another round of review:

1) The changes to the Figure 1G caption did not make it to the revised version attached.

2) In the revised paragraph starting with “This is evident in S2-1B, where the gamma band is less distinct compared…” I would strongly recommend replacing “gamma band” with “gamma cycle”.. In neuroscience literature “gamma band” is almost exclusively used to describe power in the frequency domain, whereas this is describing a visual band that appears upon trial alignment.

3) In the revised statement you inserted “Gaussian_Noise” layer. I think the underscore should be removed as this seems like a typo.

**Have the authors made all data and (if applicable) computational code underlying the findings in their manuscript fully available?**

Reviewer #1: Yes

Reviewer #2: Yes

PLOS authors have the option to publish the peer review history of their article (what does this mean?). If published, this will include your full peer review and any attached files.

Reviewer #1: **Yes: **Chunzhi Yi

Reviewer #2: **Yes: **Nicholas Tolley

---

## [Editor Report · Acceptance letter]

PCOMPBIOL-D-24-01889R2

Isolating Single Cycles of Neural Oscillations in Population Spiking

Dear Dr Sabri,

I am pleased to inform you that your manuscript has been formally accepted for publication in PLOS Computational Biology. Your manuscript is now with our production department and you will be notified of the publication date in due course.

With kind regards,

Livia Horvath
